# Graph Convolutional Policy Network for Goal-Directed Molecular Graph Generation

**Jiaxuan You**[1]*
jiaxuan@stanford.edu

**Bowen Liu**[2]*
liubowen@stanford.edu

**Rex Ying**[1]
rexying@stanford.edu

**Vijay Pande**[3]
pande@stanford.edu

**Jure Leskovec**[1]
jure@cs.stanford.edu

[1]Department of Computer Science, [2]Department of Chemistry, [3]Department of Bioengineering
Stanford University
Stanford, CA, 94305

## Abstract

Generating novel graph structures that optimize given objectives while obeying some given underlying rules is fundamental for chemistry, biology and social science research. This is especially important in the task of molecular graph generation, whose goal is to discover novel molecules with desired properties such as drug-likeness and synthetic accessibility, while obeying physical laws such as chemical valency. However, designing models to find molecules that optimize desired properties while incorporating highly complex and non-differentiable rules remains to be a challenging task. Here we propose Graph Convolutional Policy Network (GCPN), a general graph convolutional network based model for goal-directed graph generation through reinforcement learning. The model is trained to optimize domain-specific rewards and adversarial loss through policy gradient, and acts in an environment that incorporates domain-specific rules. Experimental results show that GCPN can achieve $61\%$ improvement on chemical property optimization over state-of-the-art baselines while resembling known molecules, and achieve $184\%$ improvement on the constrained property optimization task.

## 1 Introduction

Many important problems in drug discovery and material science are based on the principle of designing molecular structures with specific desired properties. However, this remains a challenging task due to the large size of chemical space. For example, the range of drug-like molecules has been estimated to be between $10^{23}$ and $10^{60}$ [32]. Additionally, chemical space is discrete, and molecular properties are highly sensitive to small changes in the molecular structure [21]. An increase in the effectiveness of the design of new molecules with application-driven goals would significantly accelerate developments in novel medicines and materials.

Recently, there has been significant advances in applying deep learning models to molecule generation [15, 38, 7, 9, 22, 4, 31, 27, 34, 42]. However, the generation of novel and valid molecular graphs that can directly optimize various desired physical, chemical and biological property objectives remains to be a challenging task, since these property objectives are highly complex [37] and non-differentiable. Furthermore, the generation model should be able to actively explore the vast chemical space, as the distribution of the molecules that possess those desired properties does not necessarily match the distribution of molecules from existing datasets.

**Present Work**. In this work, we propose Graph Convolutional Policy Network (GCPN), an approach to generate molecules where the generation process can be guided towards specified desired objectives, while restricting the output space based on underlying chemical rules. To address the challenge of goal-directed molecule generation, we utilize and extend three ideas, namely graph representation, reinforcement learning and adversarial training, and combine them in a single unified framework. Graph representation learning is used to obtain vector representations of the state of generated graphs, adversarial loss is used as reward to incorporate prior knowledge specified by a dataset of example molecules, and the entire model is trained end-to-end in the reinforcement learning framework.

**Graph representation**. We represent molecules directly as molecular graphs, which are more robust than intermediate representations such as simplified molecular-input line-entry system (SMILES) [40], a text-based representation that is widely used in previous works [9, 22, 4, 15, 38, 27, 34]. For example, a single character perturbation in a text-based representation of a molecule can lead to significant changes to the underlying molecular structure or even invalidate it [30]. Additionally, partially generated molecular graphs can be interpreted as substructures, whereas partially generated text representations in many cases are not meaningful. As a result, we can perform chemical checks, such as valency checks, on a partially generated molecule when it is represented as a graph, but not when it is represented as a text sequence.

**Reinforcement learning**. A reinforcement learning approach to goal-directed molecule generation presents several advantages compared to learning a generative model over a dataset. Firstly, desired molecular properties such as drug-likeness [1, 29] and molecule constraints such as valency are complex and non-differentiable, thus they cannot be directly incorporated into the objective function of graph generative models. In contrast, reinforcement learning is capable of directly representing hard constraints and desired properties through the design of environment dynamics and reward function. Secondly, reinforcement learning allows active exploration of the molecule space beyond samples in a dataset. Alternative deep generative model approaches [9, 22, 4, 16] show promising results on reconstructing given molecules, but their exploration ability is restricted by the training dataset.

**Adversarial training**. Incorporating prior knowledge specified by a dataset of example molecules is crucial for molecule generation. For example, a drug molecule is usually relatively stable in physiological conditions, non toxic, and possesses certain physiochemical properties [28]. Although it is possible to hand code the rules or train a predictor for one of the properties, precisely representing the combination of these properties is extremely challenging. Adversarial training addresses the challenge through a learnable discriminator adversarially trained with a generator [10]. After the training converges, the discriminator implicitly incorporates the information of a given dataset and guides the training of the generator.

GCPN is designed as a reinforcement learning agent (RL agent) that operates within a chemistry-aware graph generation environment. A molecule is successively constructed by either connecting a new substructure or an atom with an existing molecular graph or adding a bond to connect existing atoms. GCPN predicts the action of the bond addition, and is trained via policy gradient to optimize a reward composed of molecular property objectives and adversarial loss. The adversarial loss is provided by a graph convolutional network [20, 5] based discriminator trained jointly on a dataset of example molecules. Overall, this approach allows direct optimization of application-specific objectives, while ensuring that the generated molecules are realistic and satisfy chemical rules.

We evaluate GCPN in three distinct molecule generation tasks that are relevant to drug discovery and materials science: molecule property optimization, property targeting and conditional property optimization. We use the ZINC dataset [14] to provide GCPN with example molecules, and train the policy network to generate molecules with high property score, molecules with a pre-specified range of target property score, or molecules containing a specific substructure while having high property score. In all tasks, GCPN achieves state-of-the-art results. GCPN generates molecules with property scores $61\%$ higher than the best baseline method, and outperforms the baseline models in the constrained optimization setting by $184\%$ on average.

## 2   Related Work

Yang et al. [42] and Olivecrona et al. [31] proposed a recurrent neural network (RNN) SMILES string generator with molecular properties as objective that is optimized using Monte Carlo tree search

and policy gradient respectively. Guimaraes et al. [27] and Sanchez-Lengeling et al. [34] further included an adversarial loss to the reinforcement learning reward to enforce similarity to a given molecule dataset. In contrast, instead of using a text-based molecular representation, our approach uses a graph-based molecular representation, which leads to many important benefits as discussed in the introduction. Jin et al. [16] proposed to use a variational autoencoder (VAE) framework, where the molecules are represented as junction trees of small clusters of atoms. This approach can only indirectly optimize molecular properties in the learned latent embedding space before decoding to a molecule, whereas our approach can directly optimize molecular properties of the molecular graphs. You et al. [43] used an auto-regressive model to maximize the likelihood of the graph generation process, but it cannot be used to generate attributed graphs. Li et al. [25] and Li et al. [26] described sequential graph generation models where conditioning labels can be incorporated to generate molecules whose molecular properties are close to specified target scores. However, these approaches are also unable to directly perform optimization on desired molecular properties. Overall, modeling the goal-directed graph generation task in a reinforcement learning framework is still largely unexplored.

# 3 Proposed Method

In this section we formulate the problem of graph generation as learning an RL agent that iteratively adds substructures and edges to the molecular graph in a chemistry-aware environment. We describe the problem definition, the environment design, and the Graph Convolutional Policy Network that predicts a distribution of actions which are used to update the graph being generated.

## 3.1 Problem Definition

We represent a graph $G$ as $(A, E, F)$, where $A \in \{0, 1\}^{n \times n}$ is the adjacency matrix, and $F \in \mathbb{R}^{n \times d}$ is the node feature matrix assuming each node has $d$ features. We define $E \in \{0, 1\}^{b \times n \times n}$ to be the (discrete) edge-conditioned adjacency tensor, assuming there are $b$ possible edge types. $E_{i,j,k} = 1$ if there exists an edge of type $i$ between nodes $j$ and $k$, and $A = \sum_{i=1}^{b} E_i$. Our primary objective is to generate graphs that maximize a given property function $S(G) \in \mathbb{R}$, i.e., maximize $\mathbb{E}_{G'}[S(G')]$, where $G'$ is the generated graph, and $S$ could be one or multiple domain-specific statistics of interest.

It is also of practical importance to constrain our model with two main sources of prior knowledge. (1) Generated graphs need to satisfy a set of hard constraints. (2) We provide the model with a set of example graphs $G \sim p_{\text{data}}(G)$, and would like to incorporate such prior knowledge by regularizing the property optimization objective with $\mathbb{E}_{G,G'}[J(G, G')]$ under distance metric $J(\cdot, \cdot)$. In the case of molecule generation, the set of hard constraints is described by chemical valency while the distance metric is an adversarially trained discriminator.

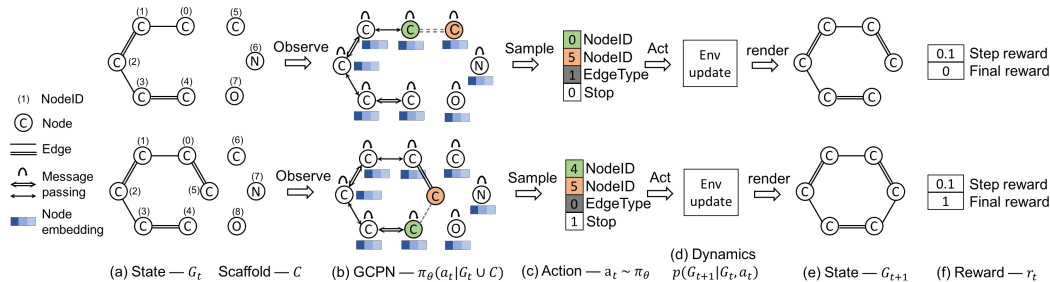

Figure 1: An overview of the proposed iterative graph generation method. Each row corresponds to one step in the generation process. **(a)** The state is defined as the intermediate graph $G_t$, and the set of scaffold subgraphs defined as $C$ is appended for GCPN calculation. **(b)** GCPN conducts message passing to encode the state as node embeddings then produce a policy $\pi_\theta$. **(c)** An action $a_t$ with 4 components is sampled from the policy. **(d)** The environment performs a chemical valency check on the intermediate state, and then returns **(e)** the next state $G_{t+1}$ and **(f)** the associated reward $r_t$.

## 3.2 Graph Generation as Markov Decision Process

A key task for building our model is to specify a generation procedure. We designed an iterative graph generation process and formulated it as a general decision process $M = (\mathcal{S}, \mathcal{A}, P, R, \gamma)$, where $\mathcal{S} = \{s_i\}$ is the set of states that consists of all possible intermediate and final graphs, $\mathcal{A} = \{a_i\}$ is the set of actions that describe the modification made to current graph at each time step, $P$ is the transition dynamics that specifies the possible outcomes of carrying out an action, $p(s_{t+1}|s_t, ...s_0, a_t)$. $R(s_t)$ is a reward function that specifies the reward after reaching state $s_t$, and $\gamma$ is the discount factor. The procedure to generate a graph can then be described by a trajectory $(s_0, a_0, r_0, ..., s_n, a_n, r_n)$, where $s_n$ is the final generated graph. The modification of a graph at each time step can be viewed as a state transition distribution: $p(s_{t+1}|s_t, ..., s_0) = \sum_{a_t} p(a_t|s_t, ...s_0)p(s_{t+1}|s_t, ...s_0, a_t)$, where $p(a_t|s_t, ...s_0)$ is usually represented as a policy network $\pi_\theta$.

Recent graph generation models add nodes and edges based on the full trajectory $(s_t, ..., s_0)$ of the graph generation procedure [43, 25] using recurrent units, which tends to "forget" initial steps of generation quickly. In contrast, we design a graph generation procedure that can be formulated as a Markov Decision Process (MDP), which requires the state transition dynamics to satisfy the Markov property: $p(s_{t+1}|s_t, ...s_0) = p(s_{t+1}|s_t)$. Under this property, the policy network only needs the intermediate graph state $s_t$ to derive an action (see Section 3.4). The action is used by the environment to update the intermediate graph being generated (see Section 3.3).

## 3.3 Molecule Generation Environment

In this section we discuss the setup of molecule generation environment. On a high level, the environment builds up a molecular graph step by step through a sequence of bond or substructure addition actions given by GCPN. Figure 1 illustrates the 5 main components that come into play in each step, namely state representation, policy network, action, state transition dynamics and reward. Note that this environment can be easily extended to graph generation tasks in other settings.

**State Space**. We define the state of the environment $s_t$ as the intermediate generated graph $G_t$ at time step $t$, which is fully observable by the RL agent. Figure 1 (a)(e) depicts the partially generated molecule state before and after an action is taken. At the start of generation, we assume $G_0$ contains a single node that represents a carbon atom.

**Action Space**. In our framework, we define a distinct, fixed-dimension and homogeneous action space amenable to reinforcement learning. We design an action analogous to link prediction, which is a well studied realm in network science. We first define a set of scaffold subgraphs $\{C_1, ..., C_s\}$ to be added during graph generation and the collection is defined as $C = \bigcup_{i=1}^s C_i$. Given a graph $G_t$ at step $t$, we define the corresponding extended graph as $G_t \cup C$. Under this definition, an action can either correspond to connecting a new subgraph $C_i$ to a node in $G_t$ or connecting existing nodes within graph $G_t$. Once an action is taken, the remaining disconnected scaffold subgraphs are removed. In our implementation, we adopt the most fine-grained version where $\mathcal{C}$ consists of all $b$ different single node graphs, where $b$ denotes the number of different atom types. Note that $\mathcal{C}$ can be extended to contain molecule substructure scaffolds [16], which allows specification of preferred substructures at the cost of model flexibility. In Figure 1(b), a link is predicted between the green and yellow atoms. We will discuss in detail the link prediction algorithm in Section 3.4.

**State Transition Dynamics**. Domain-specific rules are incorporated in the state transition dynamics. The environment carries out actions that obey the given rules. Infeasible actions proposed by the policy network are rejected and the state remains unchanged. For the task of molecule generation, the environment incorporates rules of chemistry. In Figure 1(d), both actions pass the valency check, and the environment updates the (partial) molecule according to the actions. Note that unlike a text-based representation, the graph-based molecular representation enables us to perform this step-wise valency check, as it can be conducted even for incomplete molecular graphs.

**Reward design**. Both intermediate rewards and final rewards are used to guide the behaviour of the RL agent. We define the final rewards as a sum over domain-specific rewards and adversarial rewards. The domain-specific rewards consist of the (combination of) final property scores, such as octanol-water partition coefficient (logP), druglikeness (QED) [1] and molecular weight (MW). Domain-specific rewards also include penalization of unrealistic molecules according to various criteria, such as excessive steric strain and the presence of functional groups that violate ZINC functional group filters [14]. The intermediate rewards include step-wise validity rewards and

adversarial rewards. A small positive reward is assigned if the action does not violate valency rules, otherwise a small negative reward is assigned. As an example, the second row of Figure 1 shows the scenario that a termination action is taken. When the environment updates according to a terminating action, both a step reward and a final reward are given, and the generation process terminates.

To ensure that the generated molecules resemble a given set of molecules, we employ the Generative Adversarial Network (GAN) framework [10] to define the adversarial rewards $V(\pi_\theta, D_\phi)$

$$\min_\theta \max_\phi V(\pi_\theta, D_\phi) = \mathbb{E}_{x \sim p_{data}}[\log D_\phi(x)] + \mathbb{E}_{x \sim \pi_\theta}[\log D_\phi(1 - x)] \tag{1}$$

where $\pi_\theta$ is the policy network, $D_\phi$ is the discriminator network, $x$ represents an input graph, $p_{data}$ is the underlying data distribution which defined either over final graphs (for final rewards) or intermediate graphs (for intermediate rewards). However, only $D_\phi$ can be trained with stochastic gradient descent, as $x$ is a graph object that is non-differentiable with respect to parameters $\phi$. Instead, we use $-V(\pi_\theta, D_\phi)$ as an additional reward together with other rewards, and optimize the total rewards with policy gradient methods [44] (Section 3.5). The discriminator network employs the same structure of the policy network (Section 3.4) to calculate the node embeddings, which are then aggregated into a graph embedding and cast into a scalar prediction.

## 3.4  Graph Convolutional Policy Network

Having illustrated the graph generation environment, we outline the architecture of GCPN, a policy network learned by the RL agent to act in the environment. GCPN takes the intermediate graph $G_t$ and the collection of scaffold subgraphs $C$ as inputs, and outputs the action $a_t$, which predicts a new link to be added, as described in Section 3.3.

**Computing node embeddings**.  In order to perform link prediction in $G_t \cup C$, our model first computes the node embeddings of an input graph using Graph Convolutional Networks (GCN) [20, 5, 18, 36, 8], a well-studied technique that achieves state-of-the-art performance in representation learning for molecules. We use the following variant that supports the incorporation of categorical edge types. The high-level idea is to perform message passing over each edge type for a total of $L$ layers. At the $l^{\text{th}}$ layer of the GCN, we aggregate all messages from different edge types to compute the next layer node embedding $H^{(l+1)} \in \mathbb{R}^{(n+c) \times k}$, where $n$, $c$ are the sizes of $G_t$ and $C$ respectively, and $k$ is the embedding dimension. More concretely,

$$H^{(l+1)} = \text{AGG}(\text{ReLU}(\{\tilde{D}_i^{-\frac{1}{2}} \tilde{E}_i \tilde{D}_i^{-\frac{1}{2}} H^{(l)} W_i^{(l)}\}, \forall i \in (1, ..., b))) \tag{2}$$

where $E_i$ is the $i^{th}$ slice of edge-conditioned adjacency tensor $E$, and $\tilde{E}_i = E_i + I$; $\tilde{D}_i = \sum_k \tilde{E}_{ijk}$. $W_i^{(l)}$ is a trainable weight matrix for the $i^{\text{th}}$ edge type, and $H^{(l)}$ is the node representation learned in the $l^{\text{th}}$ layer. We use AGG$(\cdot)$ to denote an aggregation function that could be one of $\{\text{MEAN}, \text{MAX}, \text{SUM}, \text{CONCAT}\}$ [12]. This variant of the GCN layer allows for parallel implementation while remaining expressive for aggregating information among different edge types. We apply a $L$ layer GCN to the extended graph $G_t \cup C$ to compute the final node embedding matrix $X = H^{(L)}$.

**Action prediction**.  The link prediction based action $a_t$ at time step $t$ is a concatenation of four components: selection of two nodes, prediction of edge type, and prediction of termination. Concretely, each component is sampled according to a predicted distribution governed by Equation 3 and 4.

$$a_t = \text{CONCAT}(a_{\text{first}}, a_{\text{second}}, a_{\text{edge}}, a_{\text{stop}}) \tag{3}$$

$$
\begin{aligned}
f_{\text{first}}(s_t) &= \text{SOFTMAX}(m_f(X)), & a_{\text{first}} &\sim f_{\text{first}}(s_t) \in \{0,1\}^n \\
f_{\text{second}}(s_t) &= \text{SOFTMAX}(m_s(X_{a_{\text{first}}}, X)), & a_{\text{second}} &\sim f_{\text{second}}(s_t) \in \{0,1\}^{n+c} \\
f_{\text{edge}}(s_t) &= \text{SOFTMAX}(m_e(X_{a_{\text{first}}}, X_{a_{\text{second}}})), & a_{\text{edge}} &\sim f_{\text{edge}}(s_t) \in \{0,1\}^b \\
f_{\text{stop}}(s_t) &= \text{SOFTMAX}(m_t(\text{AGG}(X))), & a_{\text{stop}} &\sim f_{\text{stop}}(s_t) \in \{0,1\}
\end{aligned}
\tag{4}
$$

We use $m_f$ to denote a Multilayer Perceptron (MLP) that maps $Z_{0:n} \in \mathbb{R}^{n \times k}$ to a $\mathbb{R}^n$ vector, which represents the probability distribution of selecting each node. The information from the first selected node $a_{\text{first}}$ is incorporated in the selection of the second node by concatenating its embedding $Z_{a_{\text{first}}}$

with that of each node in $G_t \cup C$. The second MLP $m_s$ then maps the concatenated embedding to the probability distribution of each potential node to be selected as the second node. Note that when selecting two nodes to predict a link, the first node to select, $a_{\text{first}}$, should always belong to the currently generated graph $G_t$, whereas the second node to select, $a_{\text{second}}$, can be either from $G_t$ (forming a cycle), or from $C$ (adding a new substructure). To predict a link, $m_e$ takes $Z_{a_{\text{first}}}$ and $Z_{a_{\text{second}}}$ as inputs and maps to a categorical edge type using an MLP. Finally, the termination probability is computed by firstly aggregating the node embeddings into a graph embedding using an aggregation function AGG, and then mapping the graph embedding to a scalar using an MLP $m_t$.

### 3.5 Policy Gradient Training

Policy gradient based methods are widely adopted for optimizing policy networks. Here we adopt Proximal Policy Optimization (PPO) [35], one of the state-of-the-art policy gradient methods. The objective function of PPO is defined as follows

$$\max L^{\text{CLIP}}(\theta) = \mathbb{E}_t[\min(r_t(\theta)\hat{A}_t, \text{clip}(r_t(\theta), 1 - \epsilon, 1 + \epsilon)\hat{A}_t)], r_t(\theta) = \frac{\pi_\theta(a_t|s_t)}{\pi_{\theta_{old}}(a_t|s_t)} \quad (5)$$

where $r_t(\theta)$ is the probability ratio that is clipped to the range of $[1 - \epsilon, 1 + \epsilon]$, making the $L^{\text{CLIP}}(\theta)$ a lower bound of the conservative policy iteration objective [17], $\hat{A}_t$ is the estimated advantage function which involves a learned value function $V_\omega(\cdot)$ to reduce the variance of estimation. In GCPN, $V_\omega(\cdot)$ is an MLP that maps the graph embedding computed according to Section 3.4.

It is known that pretraining a policy network with expert policies if they are available leads to better training stability and performance [24]. In our setting, any ground truth molecule could be viewed as an expert trajectory for pretraining GCPN. This expert imitation objective can be written as $\min L^{\text{EXPERT}}(\theta) = -\log(\pi_\theta(a_t|s_t))$, where $(s_t, a_t)$ pairs are obtained from ground truth molecules. Specifically, given a molecule dataset, we randomly sample a molecular graph $G$, and randomly select one connected subgraph $G'$ of $G$ as the state $s_t$. At state $s_t$, any action that adds an atom or bond in $G \setminus G'$ can be taken in order to generate the sampled molecule. Hence, we randomly sample $a_t \in G \setminus G'$, and use the pair $(s_t, a_t)$ to supervise the expert imitation objective.

## 4 Experiments

To demonstrate effectiveness of goal-directed search for molecules with desired properties, we compare our method with state-of-the-art molecule generation algorithms in the following tasks.

**Property Optimization**. The task is to generate novel molecules whose specified molecular properties are optimized. This can be useful in many applications such as drug discovery and materials science, where the goal is to identify molecules with highly optimized properties of interest.

**Property Targeting**. The task is to generate novel molecules whose specified molecular properties are as close to the target scores as possible. This is crucial in generating virtual libraries of molecules with properties that are generally suitable for a desired application. For example, a virtual molecule library for drug discovery should have high drug-likeness and synthesizability.

**Constrained Property Optimization**. The task is to generate novel molecules whose specified molecular properties are optimized, while also containing a specified molecular substructure. This can be useful in lead optimization problems in drug discovery and materials science, where we want to make modifications to a promising lead molecule and improve its properties [2].

### 4.1 Experimental Setup

We outline our experimental setup in this section. Further details are provided in the appendix[1].

**Dataset**. For the molecule generation experiments, we utilize the ZINC250k molecule dataset [14] that contains 250,000 drug like commercially available molecules whose maximum atom number is 38. We use the dataset for both expert pretraining and adversarial training.

**Molecule environment**. We set up the molecule environment as an OpenAI Gym environment [3] using RDKit [23] and adapt it to the ZINC250k dataset. Specifically, the maximum atom number is

Table 1: Comparison of the top 3 property scores of generated molecules found by each model.

| Method | Penalized logP | | | | QED | | | |
|---|---|---|---|---|---|---|---|---|
| | 1st | 2nd | 3rd | Validity | 1st | 2nd | 3rd | Validity |
| ZINC | 4.52 | 4.30 | 4.23 | 100.0% | 0.948 | 0.948 | 0.948 | 100.0% |
| Hill Climbing | – | – | – | – | 0.838 | 0.814 | 0.814 | 100.0% |
| ORGAN | 3.63 | 3.49 | 3.44 | 0.4% | 0.896 | 0.824 | 0.820 | 2.2% |
| JT-VAE | 5.30 | 4.93 | 4.49 | 100.0% | 0.925 | 0.911 | 0.910 | 100.0% |
| GCPN | **7.98** | **7.85** | **7.80** | **100.0%** | **0.948** | **0.947** | **0.946** | **100.0%** |

set to be 38. There are 9 atom types and 3 edge types, as molecules are represented in kekulized form. For specific reward design, we linearly scale each reward component according to its importance in molecule generation from a chemistry point of view as well as the quality of generation results. When summing up all the rewards collected from a molecule generation trajectory, the range of the reward value that the model can get is $[-4, 4]$ for final chemical property reward, $[-2, 2]$ for final chemical filter reward, $[-1, 1]$ for final adversarial reward, $[-1, 1]$ for intermediate adversarial reward and $[-1, 1]$ for intermediate validity reward.

**GCPN Setup**. We use a 3-layer defined GCPN as the policy network with 64 dimensional node embedding in all hidden layers, and batch normalization [13] is applied after each layer. Another 3-layer GCN with the same architecture is used for discriminator training. We find little improvement when further adding GCN layers. We observe comparable performance among different aggregation functions and select SUM($\cdot$) for all experiments. We found both the expert pretraining and RL objective important for generating high quality molecules, thus both of them are kept throughout training. Specifically, we use PPO algorithm to train the RL objective with the default hyperparameters as we do not observe too much performance gain from tuning these hyperparameters, and the learning rate is set as 0.001. The expert pretraining objective is trained with a learning rate of 0.00025, and we do observe that adding this objective contributes to faster convergence and better performance. Both objectives are trained using Adam optimizer [19] with batch size 32.

**Baselines**. We compare our method with the following state-of-the-art baselines. Junction Tree VAE (JT-VAE) [16] is a state-of-the-art algorithm that combines graph representation and a VAE framework for generating molecular graphs, and uses Bayesian optimization over the learned latent space to search for molecules with optimized property scores. JT-VAE has been shown to outperform previous deep generative models for molecule generation, including Character-VAE [9], Grammar-VAE [22], SD-VAE [4] and GraphVAE [39]. We also compare our approach with ORGAN [27], a state-of-the-art RL-based molecule generation algorithm using a text-based representation of molecules. To demonstrate the benefits of learning-based approaches, we further implement a simple rule based model using the stochastic hill-climbing algorithm. We start with a graph containing a single atom (the same setting as GCPN), traverse all valid actions given the current state, randomly pick the next state with top 5 highest property score as long as there is improvement over the current state, and loop until reaching the maximum number of nodes. To make fair comparison across different methods, we set up the same objective functions for all methods, and run all the experiments on the same computing facilities using 32 CPU cores. We run both deep learning baselines using their released code and allow the baselines to have wall clock running time for roughly 24 hours, while our model can get the results in roughly 8 hours.

## 4.2 Molecule Generation Results

**Property optimization**. In this task, we focus on generating molecules with the highest possible penalized logP [22] and QED [1] scores. Penalized logP is a logP score that also accounts for ring size and synthetic accessibility [6], while QED is an indicator of drug-likeness. Note that both scores are calculated from empirical prediction models whose parameters are estimated from related datasets [41, 1], and these scores are widely used in previous molecule generation papers [9, 22, 4, 39, 27]. Penalized logP has an unbounded range, while the QED has a range of $[0, 1]$ by definition, thus directly comparing the percentage improvement of QED may not be meaningful. We adopt the same evaluation method in previous approaches [22, 4, 16], reporting the best 3 property scores found by

Table 2: Comparison of the effectiveness of property targeting task.

| Method | $-2.5 \leq \text{logP} \leq -2$ | | $5 \leq \text{logP} \leq 5.5$ | | $150 \leq \text{MW} \leq 200$ | | $500 \leq \text{MW} \leq 550$ | |
|--------|---------|-----------|---------|-----------|---------|-----------|---------|-----------|
| | Success | Diversity | Success | Diversity | Success | Diversity | Success | Diversity |
| ZINC | 0.3% | 0.919 | 1.3% | 0.909 | 1.7% | 0.938 | 0 | – |
| JT-VAE | 11.3% | **0.846** | 7.6% | 0.907 | 0.7% | 0.824 | 16.0% | 0.898 |
| ORGAN | 0 | – | 0.2% | **0.909** | 15.1% | 0.759 | 0.1% | 0.907 |
| GCPN | **85.5%** | 0.392 | **54.7%** | 0.855 | **76.1%** | **0.921** | **74.1%** | **0.920** |

Table 3: Comparison of the performance in the constrained optimization task.

| $\delta$ | JT-VAE | | | GCPN | | |
|------|-------------|-------------|---------|-------------|-------------|---------|
| | Improvement | Similarity | Success | Improvement | Similarity | Success |
| 0.0 | $1.91 \pm 2.04$ | $0.28 \pm 0.15$ | 97.5% | $\mathbf{4.20 \pm 1.28}$ | $\mathbf{0.32 \pm 0.12}$ | **100.0%** |
| 0.2 | $1.68 \pm 1.85$ | $0.33 \pm 0.13$ | 97.1% | $\mathbf{4.12 \pm 1.19}$ | $\mathbf{0.34 \pm 0.11}$ | **100.0%** |
| 0.4 | $0.84 \pm 1.45$ | $\mathbf{0.51 \pm 0.10}$ | 83.6% | $\mathbf{2.49 \pm 1.30}$ | $0.47 \pm 0.08$ | **100.0%** |
| 0.6 | $0.21 \pm 0.71$ | $0.69 \pm 0.06$ | 46.4% | $\mathbf{0.79 \pm 0.63}$ | $\mathbf{0.68 \pm 0.08}$ | **100.0%** |

each model and the fraction of molecules that satisfy chemical validity. Table 1 summarizes the best property scores of molecules found by each model, and the statistics in ZINC250k is also shown for comparison. Our method consistently performs significantly better than previous methods when optimizing penalized logP, achieving an average improvement of $61\%$ compared to JT-VAE, and $186\%$ compared to ORGAN. Our method outperforms all the baselines in the QED optimization task as well, and significantly outperforms the stochastic hill climbing baseline.

Compared with ORGAN, our model can achieve a perfect validity ratio due to the molecular graph representation that allows for step-wise chemical valency check. Compared to JT-VAE, our model can reach a much higher score owing to the fact that RL allows for direct optimization of a given property score and is able to easily extrapolate beyond the given dataset. Visualizations of generated molecules with optimized logP and QED scores are displayed in Figure 2(a) and (b) respectively.

Although most generated molecules are realistic, in some very rare cases, especially where we reduce the of the adversarial reward and expert pretraining components, our method can generate undesirable molecules with astonishingly high penalized logP predicted by the empirical model, such as the one on the bottom-right of Figure 2(a) in which our method correctly identified that Iodine has the highest per atom contribution in the empirical model used to calculate logP. These undesirable molecules are likely to have inaccurate predicted properties and illustrate an issue with optimizing properties calculated by an empirical model, such as penalized logP and QED, without incorporating prior knowledge. Empirical prediction models that predict molecular properties generalize poorly for molecules that are significantly different from the set of molecules used to train the model. Without any restrictions on the generated molecules, an optimization algorithm would exploit the lack of generalizability of the empirical property prediction models in certain areas of molecule space. Our model addresses this issue by incorporating prior knowledge of known realistic molecules using adversarial training and expert pretraining, which results in more realistic molecules, but with lower property scores calculated by the empirical prediction models. Note that the hill climbing baseline algorithm mostly generates undesirable cases where the accuracy of the empirical prediction model is questionable, thus its performance with optimizing penalized logP is not listed on Table 1.

**Property Targeting**. In this task, we specify a target range for molecular weight (MW) and logP, and report the percentage of generated molecules with property scores within the range, as well as the diversity of generated molecules. The diversity of a set of molecules is defined as the average pairwise Tanimoto distance between the Morgan fingerprints [33] of the molecules. The RL reward for this task is the L1 distance between the property score of a generated molecule and the range center. To increase the difficulty, we set the target range such that few molecules in ZINC250k dataset are within the range to test the extrapolation ability of the methods to optimize for a given target. The target ranges include $-2.5 \leq \text{logP} \leq -2$, $5 \leq \text{logP} \leq 5.5$, $150 \leq \text{MW} \leq 200$ and $500 \leq \text{MW} \leq 550$.

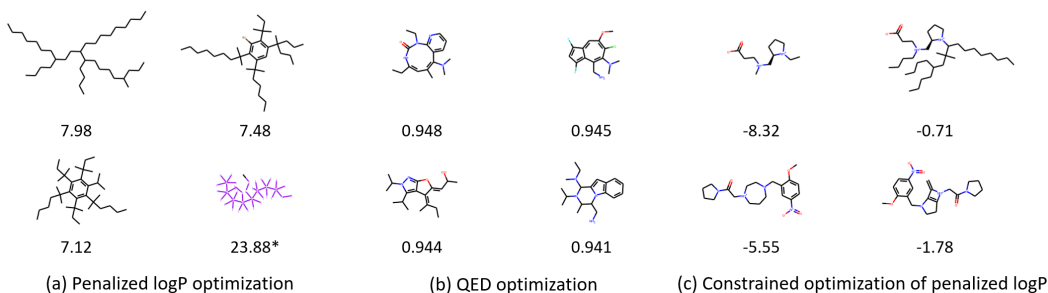

| | | | |
|---|---|---|---|
| 7.98 | 7.48 | 0.948 | 0.945 | -8.32 | -0.71 |
| 7.12 | 23.88* | 0.944 | 0.941 | -5.55 | -1.78 |
| (a) Penalized logP optimization | | (b) QED optimization | | (c) Constrained optimization of penalized logP | |

Figure 2: Samples of generated molecules in property optimization and constrained property optimization task. In **(c)**, the two columns correspond to molecules before and after modification.

As is shown in Table 2, GCPN has a significantly higher success rate in generating molecules with properties within the target range, compared to baseline methods. In addition, GCPN is able to generate molecules with high diversity, indicating that it is capable of learning a general stochastic policy to generate molecular graphs that fulfill the property requirements.

**Constrained Property Optimization**. In this experiment, we optimize the penalized logP while constraining the generated molecules to contain one of the 800 ZINC molecules with low penalized logP, following the evaluation in JT-VAE. Since JT-VAE cannot constrain the generated molecule to have certain structure, we adopt their evaluation method where the constraint is relaxed such that the molecule similarity $sim(G, G')$ between the original and modified molecules is above a threshold $\delta$.

We train a fixed GCPN in an environment whose initial state is randomly set to be one of the 800 ZINC molecules, then conduct the same training procedure as the property optimization task. Over the 800 molecules, the mean and standard deviation of the highest property score improvement and the corresponding similarity between the original and modified molecules are reported in Table 3. Our model significantly outperforms JT-VAE with 184% higher penalized logP improvement on average, and consistently succeeds in discovering molecules with higher logP scores. Also note that JT-VAE performs optimization steps for each given molecule constraint. In contrast, GCPN can generalize well: it learns a general policy to improve property scores, and applies the same policy to all 800 molecules. Figure 2(c) shows that GCPN can modify ZINC molecules to achieve high penalized logP score while still containing the substructure of the original molecule.

# 5   Conclusion

We introduced GCPN, a graph generation policy network using graph state representation and adversarial training, and applied it to the task of goal-directed molecular graph generation. GCPN consistently outperforms other state-of-the-art approaches in the tasks of molecular property optimization and targeting, and at the same time, maintains $100\%$ validity and resemblance to realistic molecules. Furthermore, the application of GCPN can extend well beyond molecule generation. The algorithm can be applied to generate graphs in many contexts, such as electric circuits, social networks, and explore graphs that can optimize certain domain specific properties.

# 6   Acknowledgements

The authors thank Xiang Ren, Marinka Zitnik, Jiaming Song, Joseph Gomes, Amir Barati Farimani, Peter Eastman, Franklin Lee, Zhenqin Wu and Paul Wender for their helpful discussions. This research has been supported in part by DARPA SIMPLEX, ARO MURI, Stanford Data Science Initiative, Huawei, JD, and Chan Zuckerberg Biohub. The Pande Group acknowledges the generous support of Dr. Anders G. Frøseth and Mr. Christian Sundt for our work on machine learning. The Pande Group is broadly supported by grants from the NIH (R01 GM062868 and U19 AI109662) as well as gift funds and contributions from Folding@home donors.

V.S.P. is a consultant & SAB member of Schrodinger, LLC and Globavir, sits on the Board of Directors of Apeel Inc, Asimov Inc, BioAge Labs, Freenome Inc, Omada Health, Patient Ping, Rigetti Computing, and is a General Partner at Andreessen Horowitz.

# 7 Appendix

**Validity**. We define a molecule as valid if it is able to pass the sanitization checks in RDKit.

**Valency**. This specifies the chemically allowable node degrees for an atom of a particular element. Some elements can have multiple possible valencies. At each intermediate step, the molecule environment checks that each atom in the partially completed graph has not exceeded its maximum possible valency of that element type.

**Steric strain filter**. Valid molecules can still be unrealistic. In particular, it is possible to generate valid molecules that are very sterically strained such that it is unlikely that they will be stable at ordinary conditions. We designed a simple steric strain filter that performs MMFF94 forcefield [11] minimization on a molecule, and then penalizes it as being too sterically strained if the average angle bend energy exceeds a cutoff of 0.82 kcal/mol.

**Reactive functional group filter**. We also penalize molecules that possess known problematic and reactive functional groups. For simplicity, we use the same set of rules that was used in the construction of the ZINC dataset, as implemented in RDKit.

**Reward design implementation**. For property optimization task, we use linear functions to roughly map the minimum and maximum property score of ZINC dataset into the desired reward range. For property targeting task, we use linear functions to map the absolute difference between the target and the property score into the desired reward range. We threshold the reward such that the reward will not exceed the desired reward range, as is described in Section 4.1. For specific parameters, please refer to the open-sourced code: `https://github.com/bowenliu16/rl_graph_generation`

## Footnotes

*The two first authors made equal contributions.

[1]Link to code and datasets: `https://github.com/bowenliu16/rl_graph_generation`

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
