[Reviews · NeurIPS 2018]

Reviewer 1



In this paper, the authors consider the problem of molecular graph generation and propose GCPN, a generative model based on GCN, RL and Adversarial-training. The proposed model takes turns to compute node embedding of current graph and predict actions to link nodes or halt. It is able to generate chemical molecule with 100% chemical validity and achieves much better performances compared to most recent state-of-the-art. Quality & Significance: 7. The paper is overall of high quality. The experimental results are very impressive especially given the recent state-of-the-art JTVAE has already achieved significant improvement compared to previously proposed models. However, as RL algorithms are known to be hard to tune, I would suggest the authors release their code soon. Clarity: 9. This paper is clearly written and very easy to follow and understand. Originality: 6. There has been attempts for using RL to guide the generation of chemical molecule, and this paper takes one step further to make it work on graph representations. The paper is a strong step in the topic of molecular graph generation and I would suggest an accept.

Reviewer 2



This paper proposed a graph generation algorithm based on reinforcement learning. A graph convolutional neural network is used to encode the node representations, which are further used to perform action prediction. Policy gradient algorithm is used to learn the optimal policy function. Experimental results on the task of molecule generation prove the effectiveness of the proposed algorithm over several existing state-of-the-art algorithms including JT-VAE and ORGAN. Strength: - the paper is well written and easy to follow - the proposed algorithm is simple and elegant Weakness: - some details are missing. For example, how to design the rewards is not fully understandable. - some model settings are arbitrarily set and are not well tested. For example, what is the sensitivity of the model performance w.r.t. the number of layers used in GCN for both the generator and discriminator?

Reviewer 3



The paper proposes combining graph neural nets, RL, and adversarial learning to train a molecule generation procedure optimized according to specific task requirements. RL is used to optimize non-differentiable objectives. Adversarial learning is used to encourage the policy to generate molecules that are similar to known ones. Graph neural nets are used to capture the graph structure of the molecules. Results are shown for three different tasks -- optimizing a specific property of the generated molecule, generating molecules that satisfy constraints on property scores, and generating molecules that satisfy constraints on their composition. ZINC dataset is used for training. Results show the proposed approach outperforming other learning based approaches. Strengths: - The paper is well-written and easy to read. - Although the proposed approach is a straightforward combination of existing ideas, it is still interesting to explore the performance of such a combination on the kind of tasks considered in the paper. Weaknesses: - In the evaluation, there is no comparison to simple baselines with no learning, like Genetic Algorithms, Hill Climbing, random search, etc. which makes it difficult to evaluate how much benefit does sophisticated learning-based approaches provide. - In the optimization tasks, the running time, computational budget, and the number of function evaluations should all be controlled across the various algorithms (at least approximately) so that the comparisons are fair. There is no mention of these factors in the paper. Since each optimization algorithm represents a trade-off between solution quality and time/computation/evaluation budget, it is not clear where on that trade-off curve each algorithm being compared is and whether they are even comparable. (If one only considers solution quality, even brute force search will do well.) This makes it hard to decide whether the results are interesting. Comments/Questions: - How are the potentially unstable learning dynamics of adversarial learning and RL handled? Insights into the challenges can be very useful. - Lines 171-172 mention using "small" positive and negative rewards to discourage infeasible actions. What value is determined to be "small" and how is it determined? - If the policy becomes deterministic, then it will always generate the same molecule. How is the policy kept stochastic so that it generates diverse molecules? - There is a missing reference in the Supplementary Material.